# Efficacy and safety of ultrasound-guided radiofrequency ablation combined with transhepatic artery embolization chemotherapy for hepatocellular carcinoma: A meta-analysis

**Kerui Pan, Sisi Wang, Xueping Li, Shuoming Wu** *

Lianyungang First People's Hospital, China

* withyou31@163.com

## Abstract

### Objective

Meta-analysis was used to assess the efficacy and safety of ultrasound-guided radiofrequency ablation combined with transhepatic artery embolization chemotherapy for hepatocellular carcinoma.

### Methods

Randomized controlled studies on ultrasound-guided radiofrequency ablation combined with transhepatic artery embolization chemotherapy for hepatocellular carcinoma were searched in the databases of PubMed, Embase, Cochrane library, web of science with a search deadline of March 14, 2024. Data were analyzed using Stata 15.0.

### Result

Six randomized controlled studies involving 520 individuals were finally included, the results of meta-analysis showed that ultrasound-guided radiofrequency ablation combined with TACE can improve objective response rate [RR = 1.52, 95%CI (1.28, 1.81)], improve disease control rate [RR = 1.15, 95%CI (1.06, 1.24)], The survival rate [RR = 1.34, 95%CI (1.19,1.51)] did not increase adverse reactions [RR = 1.34, 95%CI (1.00,1.79)].

### Conclusion

Based on the findings of the current study, ultrasound-guided radiofrequency ablation combined with TACE was found to improve the objective remission rate, disease control rate, and did not increase adverse events in patients with hepatocellular carcinoma.

**Data Availability Statement:** All relevant data are within the paper and Supporting Information files.

**Funding:** The author(s) received no specific funding for this work.

**Competing interests:** The authors have declared that no competing interests exist.

## Introduction

Liver cancer is a major health problem, with more than 850,000 cases per year worldwide [1]. This tumor is currently the second leading cause of cancer-related deaths globally, and this number is rising [2, 3]. Globally, liver cancer is the sixth most common cancer and the second leading cause of cancer-related deaths (approximately 800,000 cases per year) [4]. 85–90% of all primary liver cancers are Hepatocellular carcinoma (HCC). Various risk factors for the development of HCC are well defined such as cirrhosis (regenerating nodules to differentiate cirrhosis from fibrosis), hepatitis B virus (HBV) infection, hepatitis C virus (HCV) infection, alcoholism, and metabolic syndrome [5–7]. Due to the high incidence of HCC, the economic burden on society and families is high. Therefore, a better treatment needs to be found [8]. Currently, radical surgery is the first choice for primary liver cancer treatment, but due to the lack of obvious symptoms in the early stage of the disease and the insidious progression of the disease [9], most patients are in the advanced stage of liver cancer when diagnosed and miss the best time for surgery, while transcatheter arterial chemoembolization, molecular targeted therapy, radiotherapy, radiofrequency ablation and other means are the mainstays of the treatment of this kind of patients [10–12].

Transcatheter arterial chemoembolization (TACE) blocks the arteries supplying blood to the liver cancer tissues by blocking the arterial blood supply, and then instills chemotherapeutic drugs to achieve the purpose of inhibiting and killing liver cancer cells [13], but due to the incomplete filling of the embolus agent in the tumor, the overall therapeutic effect is not satisfactory, and most of the patients with primary cancers have been treated with chemotherapy [14]. However, due to the possibility of incomplete filling of embolic agents in the tumor, the overall therapeutic effect is not satisfactory, and most patients with primary liver cancer will undergo secondary surgery, and the prognosis of patients is poor due to the inability of some patients to tolerate the secondary surgery or the high risk of the secondary surgery [14, 15]. Ultrasound plays an important role in the early diagnosis of hepatocellular carcinoma, and the application of ultrasound-guided radiofrequency ablation in hepatocellular carcinoma has gradually matured with the promotion of minimally invasive surgery, which has the characteristics of small trauma and high reproducibility [16, 17]. Ultrasound-guided radiofrequency ablation is a minimally invasive surgical method, which can directly remove the tumor lesions, and can effectively make up for the therapeutic defects of TACE by performing more accurate and safe surgical operations under ultrasound guidance [18, 19]. The efficacy of ultrasound-guided radiofrequency ablation combined with TACE in hepatocellular carcinoma is still controversial [20], so the present study hopes to resolve the controversy by meta-analysis and provide a new choice for clinical patients in the treatment.

## Method

The systematic review described herein was accepted by the online PROSPERO international prospective register of systematic reviews [21] of the National Institute for Health Research (CRD42024519464). This meta-analysis does not involve human subjects. IRB review is not required.

### Inclusion and exclusion criteria

The included population met the diagnostic criteria for hepatocellular carcinoma [22]. ultrasound-guided radiofrequency ablation combined with TACE was used in the experimental group and TACE was used in the control group, and the primary outcome were objective response rate, disease control rate, and the secondary outcome were survival and adverse event, the randomized controlled trial was included in this study.

Conference abstracts, meta-analyses, systematic reviews, animal experiments, Full text is not available and case reports, people who have previously received other treatments will be considered for exclusion.

## Literature retrieval

Randomized controlled trials on ultrasound-guided radiofrequency ablation combined with TACE versus TACE for hepatocellular carcinoma were searched in PubMed, Embase, Cochrane Library, Web of science, with a search deadline of March 14, 2024, using the mesh word combined with a free word: ultrasound, radiofrequency ablation, hepatocellular carcinoma, and TACE. Detailed search strategies are provided in S1 Table.

## Data extract

Two authors (Shuoming Wu and Sisi Wang) rigorously screened the literature based on predetermined inclusion and exclusion criteria. In case of any disagreement, they resolved it through discussion or sought the opinion of a third person (Kerui Pan) to negotiate and reach consensus. Information extracted from the included studies included the following key details: study, year, sample size, age, gender, tumor staging, child-Pugh liver function, and outcome.

## Included studies' risk of bias

Two investigators (Shuoming Wu and Sisi Wang) independently assessed the risk of bias as low, unclear, or high using the Cochrane Collaboration's tools [23]. If there was any disagreement, a third person (Kerui Pan) was consulted to reach consensus. The assessment included seven areas: generation of randomized sequences (selective bias), allocation concealment (selective bias), blinding of implementers and participants (implementation bias), blinding of outcome assessors (observational bias), completeness of outcome data (follow-up bias), selective reporting of study results (reporting bias), and other potential sources of bias. Each included study was assessed individually against these criteria. If a study fully met all criteria, it was at "low risk" of bias, indicating a high-quality study and low overall risk of bias. If a study partially met the criteria, its quality was categorized as 'unclear risk', indicating a moderate likelihood of bias. If a study did not meet the criteria at all, it was categorized as "high risk", indicating a high risk of bias and low quality of the study.

## Data analysis

The collected data were statistically analyzed using Stata 15.0 software (Stata Corp, College Station, TX, USA). Heterogeneity between included studies was assessed using I2 values or Q-statistics. I2 values of 0%, 25%, 50%, and 75% indicated no heterogeneity, low heterogeneity, moderate heterogeneity, and high heterogeneity, respectively. If the I2 value was equal to or greater than 50%, a sensitivity analysis was performed to explore potential sources of heterogeneity. If heterogeneity was less than 50 per cent, analyses were conducted using a fixed-effects model. Standardized mean difference (SMD) and 95% confidence interval (CI) were used for continuous variables and odds ratio (OR) and 95% confidence interval (CI) for dichotomous variables. In addition, random effects model and Egger's test were used to assess publication bias.

## Result

Fig 1 shows our literature search process, which initially retrieved 4422 documents, removed 1639 duplicates, removed 2767 articles by reading titles and abstracts, removed 10 papers by reading the full text, and finally included 6 randomized controlled trials [24–29] for analysis.

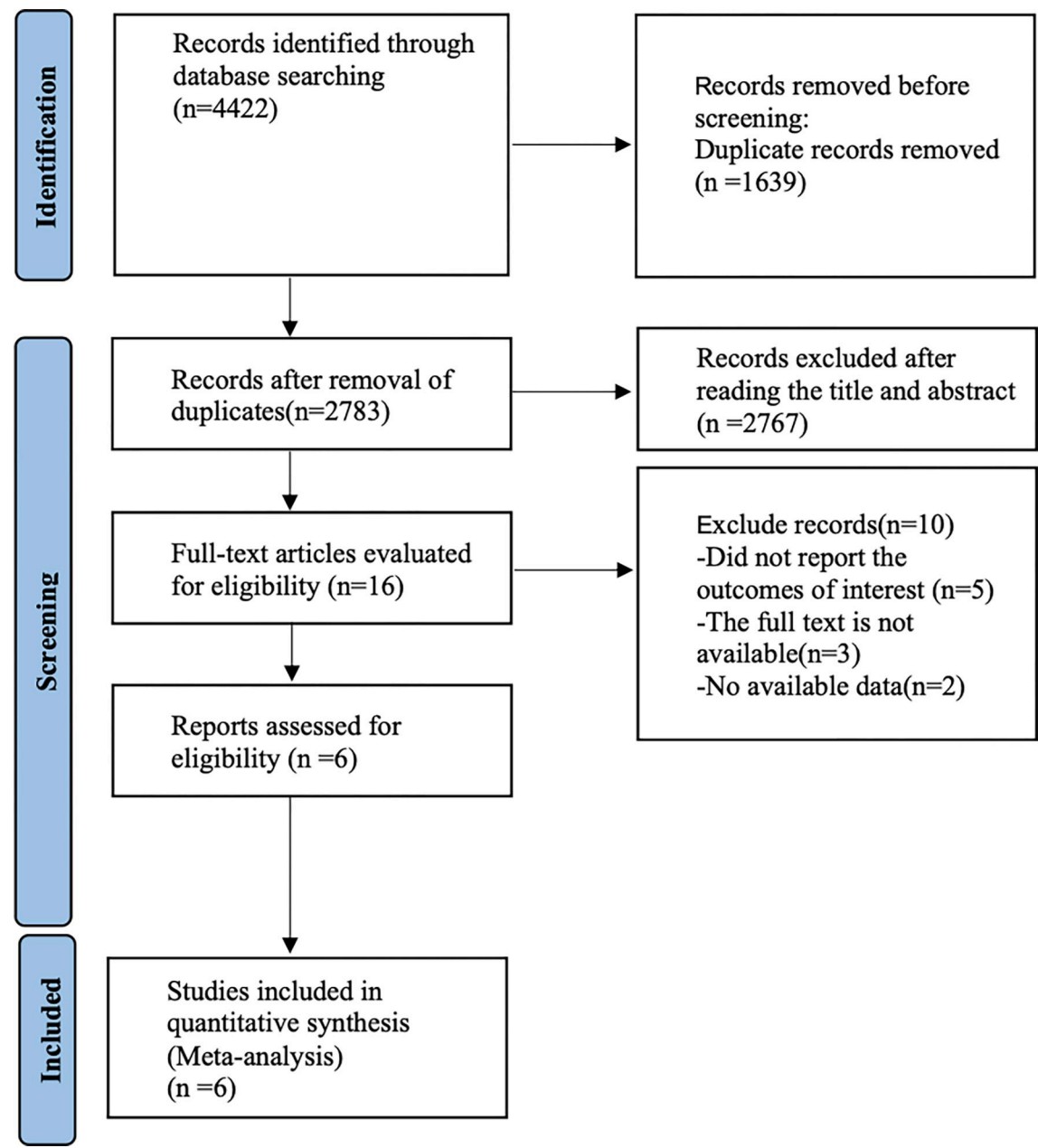

**Fig 1. Literature search flow chart.**

### Basic characteristics and risk of bias of the included studies

Six randomized controlled studies involving 520 individuals were finally included, aged 48–62 years, Baseline characteristics are shown in Table 1 The six included studies clearly accounted for the method of randomization used, and the risk of bias results are shown in Figs 2 and 3.

### Result of meta-analysis

**Objective response rate.** 5 articles mentioned the objective response rate, which was tested for heterogeneity ($I^2 = 0\%$, $P = 0.911$), therefore fixed effects model was used. The results

**Table 1. Basic features of literature.**

| Study | Year | Sample size | Gender (M/F) | Mean age (years) | WHO tumor staging | TACE drugs | Child-Pugh liver function | Outcome |
|---|---|---|---|---|---|---|---|---|
| He | 2021 | U-RFA+TACE:36 TACE:36 | 43/29 | U-RFA+TACE:52.3 TACE:52.1 | IIIB-IV | Gefitinib 250 mg/d | NR | ORR; DCR; |
| Xu | 2016 | U-RFA+TACE:38 TACE:34 | 55/17 | 50.1 | NR | Pirarubicin | A/B | OS; AEs; |
| Wang | 2018 | U-RFA+TACE:15 TACE:15 | 17/13 | U-RFA+TACE:47.6 TACE:48.7 | NR | Azithromycin, oxaliplatin, mitomycin and fluorouracil injections | A/B | ORR; DCR; OS; AEs |
| Zhu | 2018 | U-RFA+TACE:62 TACE:62 | 92/32 | U-RFA+TACE:59.2 TACE:58.5 | I-IV | Pirarubicin 20~40 mg, mitomycin 6~12 mg, oxaliplatin 60~120 mg | A/B | ORR; DCR; OS |
| Wang | 2017 | U-RFA+TACE:36 TACE:36 | 41/31 | U-RFA +TACE:52.07 TACE:52.13 | NR | 20mg pirarubicin, 150mg oxaliplatin, 8mg mitomycin | NR | ORR; DCR; AEs; |
| Li | 2023 | U-RFA+TACE:40 TACE:40 | 60/20 | U-RFA+TACE:56.5 TACE:57.1 | IV | 50-150mg oxalplatin, 30-50mg epirubicin | NR | AEs; |
| Wang | 2022 | U-RFA+TACE:35 TACE:35 | 39/31 | U-RFA +TACE:62.37 TACE:61.84 | II-III | Cisplatin, pirarubicin hydrochloride | A/B | ORR; DCR; OS; |

U-RFA: Ultrasound guided radiofrequency ablation; TACE: Transcatheter arterial chemoembolization; M/F: Male/Female; ORR: Objective response rate; DCR: Disease control rate; OS: Survival rate; AEs: adverse events

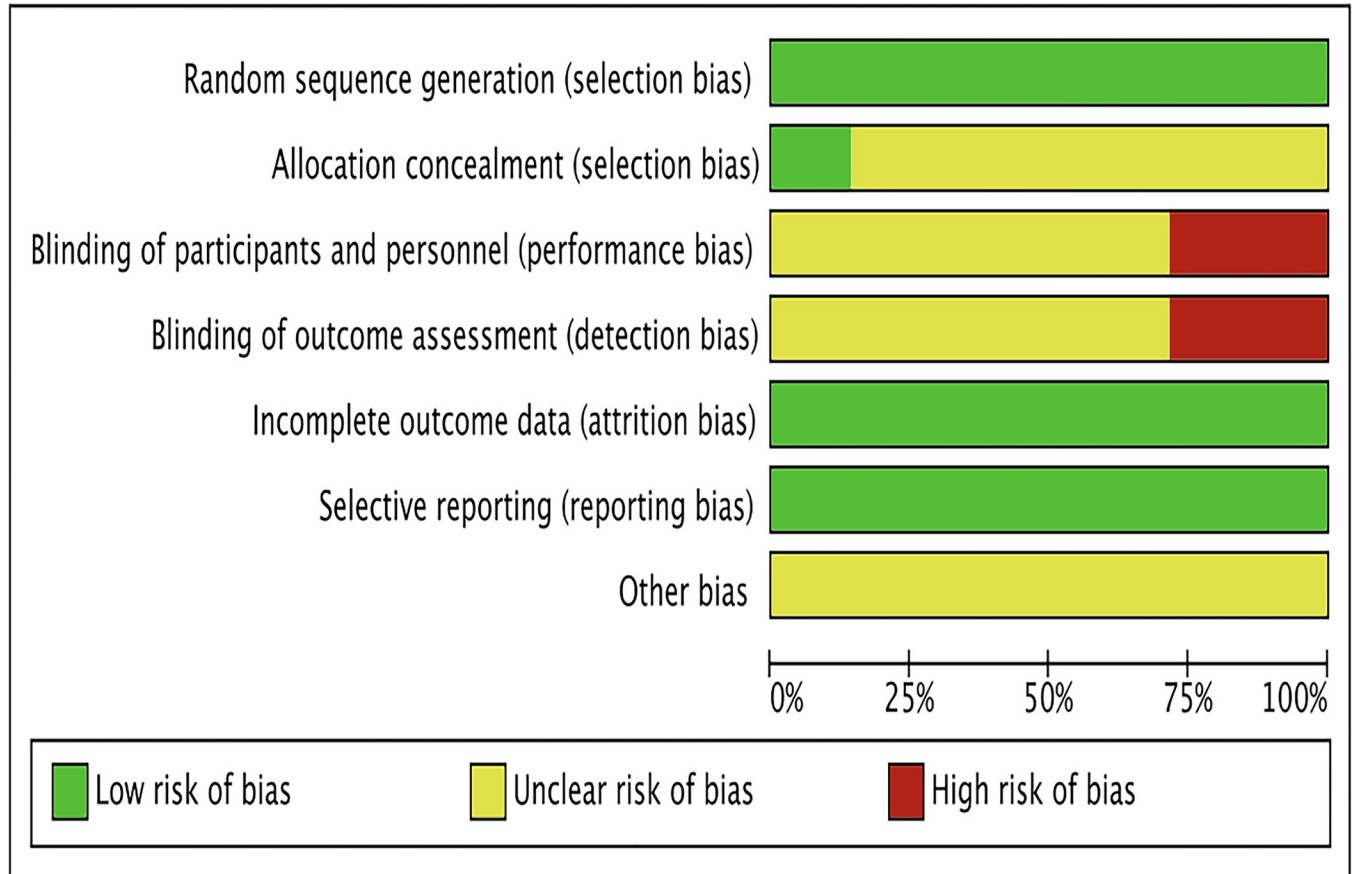

**Fig 2. Risk of bias graph.**

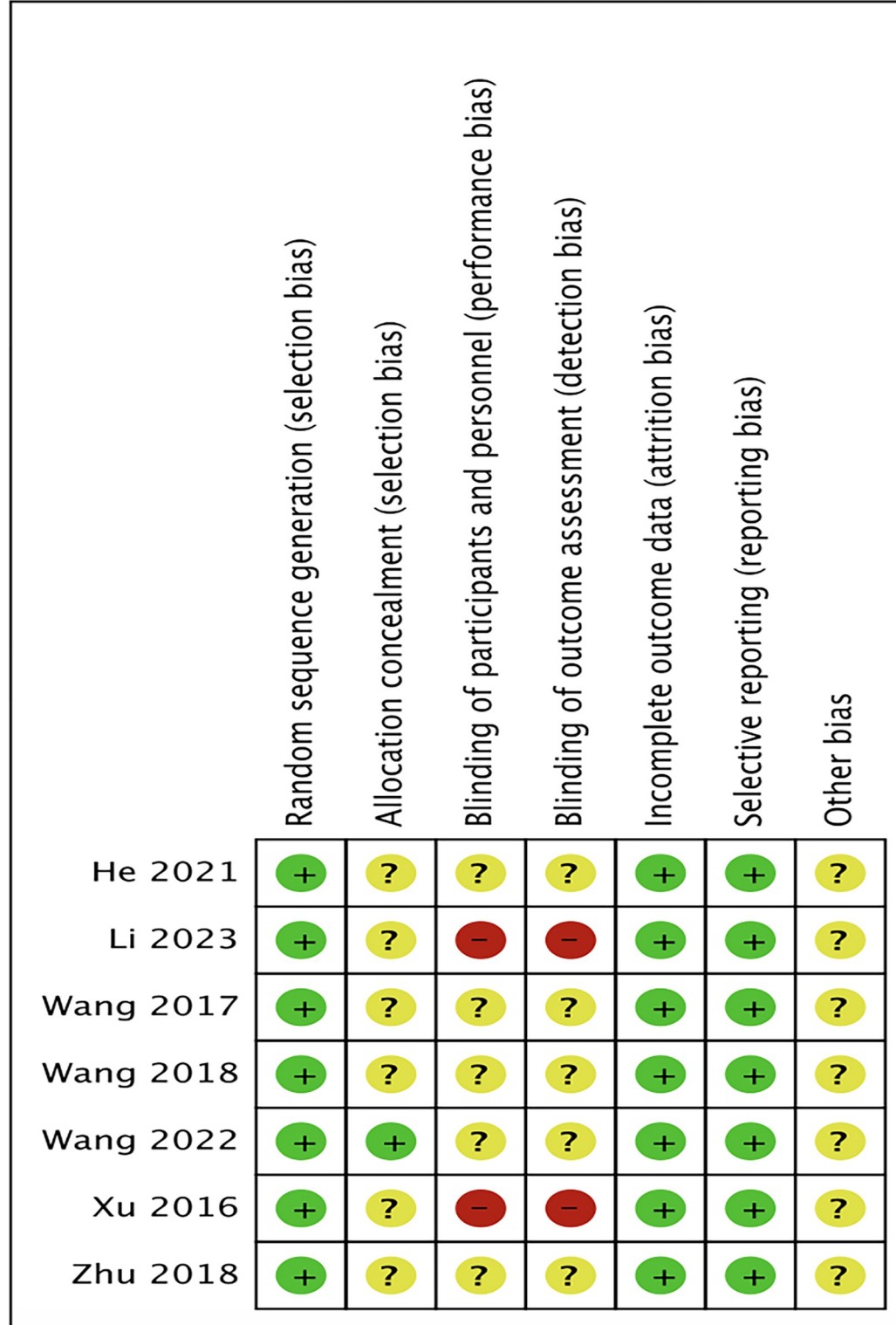

**Fig 3. Risk of bias summary.**

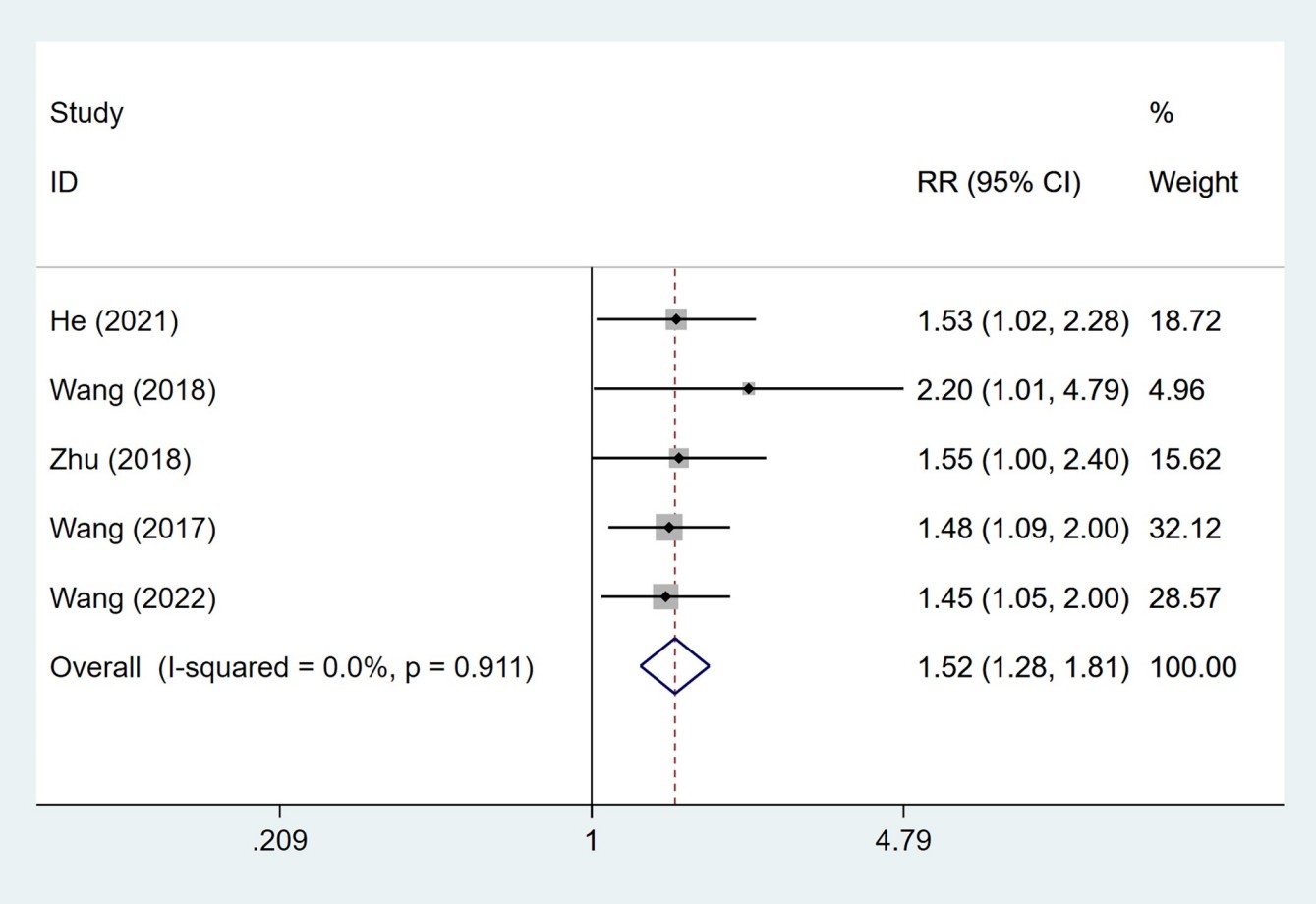

**Fig 4. Forest plot of objective response rate meta-analysis.**

of the analysis (Fig 4) suggested that ultrasound-guided radiofrequency ablation combined with TACE can improve objective response rate [RR = 1.52, 95% CI (1.28, 1.81)].

**Disease control rate.**   5 articles mentioned the disease control rate, which was tested for heterogeneity ($I^2$ = 48%, P = 0.103), therefore fixed effects model was used. The results of the analysis (Fig 5) suggested that ultrasound-guided radiofrequency ablation combined with TACE can improve disease control rate [RR = 1.15, 95% CI (1.06, 1.24)].

**Survival rate.**   4 article mentioned the survival rate, the heterogeneity test ($I^2$ = 33.8%, P = 0.137), so the fixed effect model was used for the analysis, and the results (Fig 6) of the analysis suggested that ultrasound-guided radiofrequency ablation combined with TACE can improve the survival rate [RR = 1.34,95%CI(1.19,1.51)], and we performed subgroup analyses based on survival time, in which one-year survival [RR = 1.31, 95%CI (1.13,1.50)], two-year survival [RR = 1.28, 95%CI (1.01,1.62)], three-year survival [RR = 2.12, 95%CI (1.36, 3.31)].

**Adverse events.**   4 articles mentioned adverse reactions (including fever, vomiting, and jaundice), and the test of heterogeneity ($I^2$ = 49%,P = 0.056), therefore, the analysis was performed using a fixed-effects model, and the results (Fig 7) of the analysis suggested that ultrasound-guided radiofrequency ablation combined with TACE does not increase adverse reactions [RR = 1.34, 95%CI (1.00,1.79)], we which fever [RR = 1.62, 95%CI (0.97,2.70)], vomiting [RR = 1.26, 95%CI (0.87,1.83)], jaundice [RR = 1.01, 95%CI (0.37,2.72)].

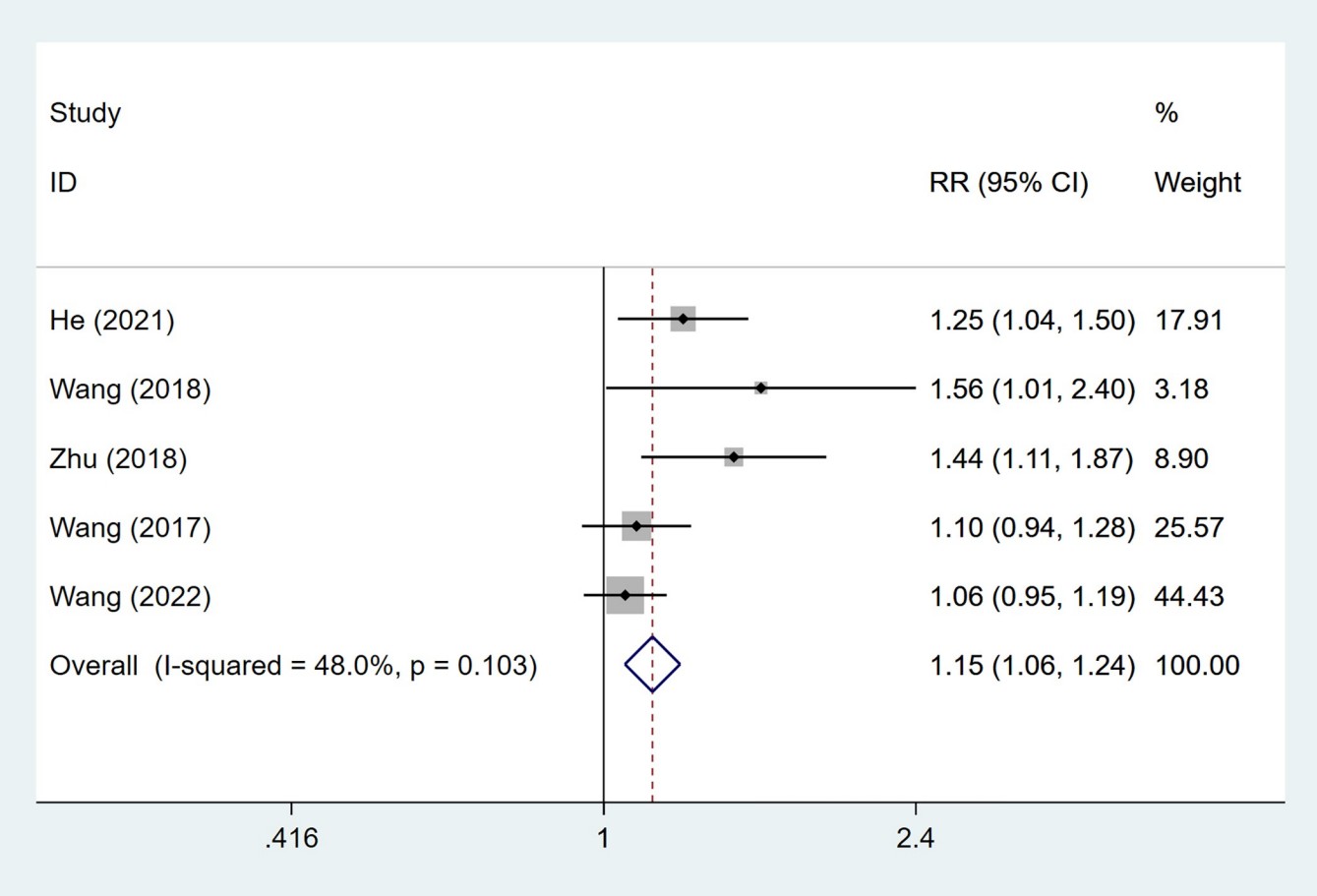

**Fig 5. Forest plot of disease control rate meta-analysis.**

**Published bias.** Publication bias was assessed by an Egger's test for objective response rate, disease control rate, survival rate, adverse reactions. Which showed no publication bias for objective response rate (p = 0.08), disease control rate (P = 0.24), survival rate (P = 0.21), adverse events (P = 0.133).

## Discussion

Previous study have investigated radiofrequency ablation combined with TACE in the treatment of hepatocellular carcinoma, but without ultrasound induction [30]; therefore, to the best of our knowledge, this is the first time to evaluate the efficacy and safety of ultrasound-guided radiofrequency ablation combined with TACE in the treatment of hepatocellular carcinoma. Ultrasound-guided radiofrequency ablation is a new therapeutic modality, the main mechanism of which is to insert radiofrequency electrodes into the tumor under the guidance of ultrasound, turn on the electrodes, emit electromagnetic waves, oscillate the tumor tissues, emit huge heat energy, distribute the tumor tissues at a temperature of 90~100°C, cleave the DNA chain of the tumor at a high temperature, denature the proteins, coagulate the blood vessels around the tumor, and thus treat the liver cancer cells and the liver cancer cells [31, 32]. This can effectively prevent the metastasis of liver cancer cells to the liver or the whole body.

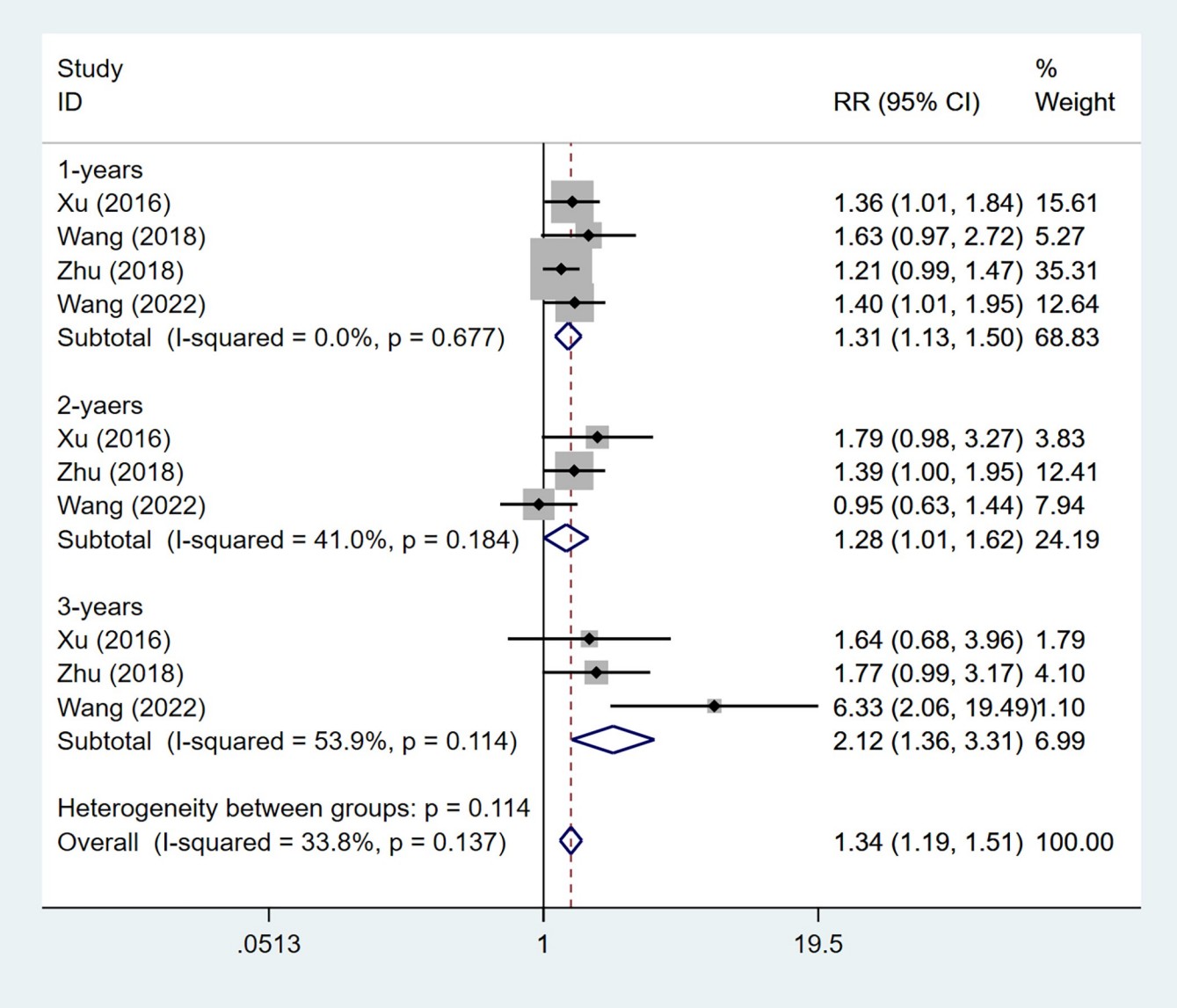

**Fig 6. Forest plot of survival rate meta-analysis.**

In our current study, we found that the ultrasound-guided radiofrequency ablation modality combined with TACE increased the objective remission rate, disease control rate, and survival rate of hepatocellular patients, and did not increase adverse events, The high survival rate and disease remission rate after combined treatment may be due to the following reasons: first, iodide precipitates around the lesion during combined treatment. Therefore, it can not only be used as a marker for radiofrequency ablation to facilitate the operator to recognize the ablation area, but also as a heat-conducting medium to improve the ablation efficiency and keep the surrounding hepatocellular carcinoma microenvironment in a static state [33, 34]. By improving the ablation effect, tumor recurrence can be reduced; second, TACE can reduce the heat loss during RF ablation by blocking blood flow into the tumor [35]; third, chemotherapeutic agents targeting malignant tumors increase the effect of high body temperature on cancer cells. Finally, TACE can further treat microscopic lesions that cannot be detected by the naked

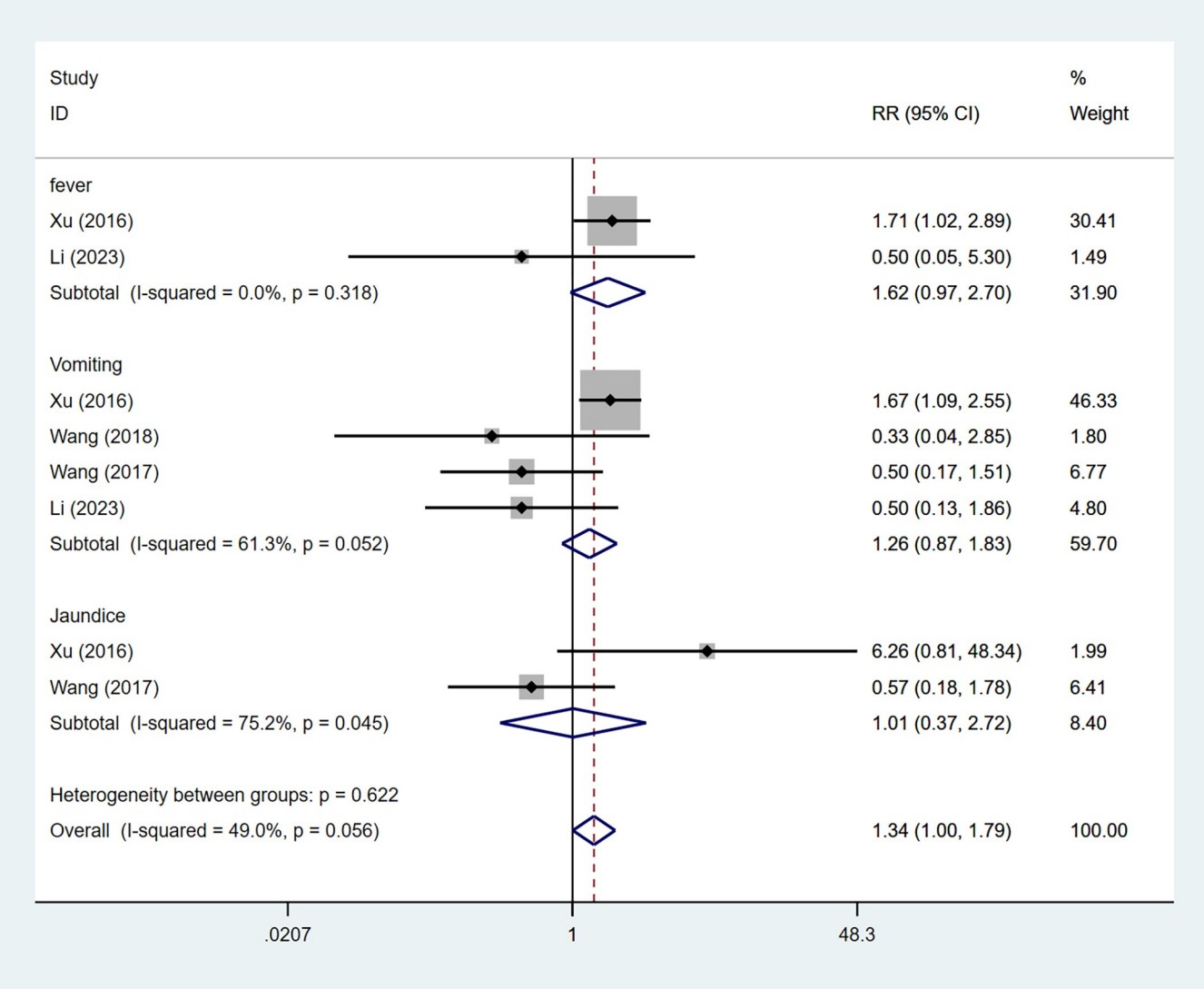

**Fig 7. Forest plot of adverse reactions meta-analysis.**

eye or imaging, thus improving patient survival and disease remission rates. According to the Barcelona Clinical Hepatocellular Carcinoma Group guidelines [36, 37], unresectable hepatocellular carcinoma outside the criteria for ablation is considered unsuitable for ablation, and palliative treatments, such as TACE, are recommended. TACE can reveal preoperatively undetected microscopic lesions, embolize blood vessels supplying the tumor, and at the same time reduce the "heat-loss effect", kill tumor cells, and reduce the size and number of tumors to achieve stage reduction [38]. The size and number of tumor cells can be reduced, and the tumor can be downstage. TACE and combined ultrasound-guided radiofrequency ablation as a treatment for hepatocellular carcinoma beyond the ablation standard has a good long-term effect. Shi et al. [39] studied the efficacy of combined ultrasound-guided radiofrequency ablation (72 cases) and ultrasound-guided radiofrequency ablation (357 cases) after downstage of TACE, and the results showed that the 1-, 3-, and 5-year OS in the combined group and the ultrasound-guided radiofrequency ablation group were 99%, 80%, 66%, and 94%, 84%, and

66%, respectively, 66% and 94%, 84%, 69%, respectively. The efficacy of TACE combined with ultrasound-guided radiofrequency ablation compared with ultrasound-guided radiofrequency ablation alone for the treatment of a single hepatocellular carcinoma of 3.1–5.0 cm in diameter has also been reported in the literature [40], which concluded that TACE combined with ultrasound-guided radiofrequency ablation for the treatment of hepatocellular carcinoma of 3.1–5.0 cm in diameter of a single node showed a better local tumor control rate and patient survival than that of ultrasound-guided radiofrequency ablation therapy alone [41].

However, our current study still has several limitations. First, the studies were all from Asian populations, using differences in Child-Pugh classification, tumor size, tumor number, and tumor stage. These factors may affect the reliability of the conclusions; second, due to the limited number of included studies, the results should be interpreted with caution; third, the inclusion of malefactors lacked detailed implementation details about the blinding and random allocation methods, which increased the risk of correlation bias.

## Conclusion

Based on the results of the current study, ultrasound-guided radiofrequency ablation combined with TACE may improve the objective remission rate and disease control rate in patients with hepatocellular carcinoma without increasing adverse events. However, due to the existence of study limitations, we hope that more high-quality randomized controlled studies will be available in the future to support our opinion.

## Supporting information

**S1 Checklist. Prisma checklist.**
(DOCX)

**S1 Table. Search history.**
(DOCX)

**S1 File. Extract data details.**
(ZIP)

## Author Contributions

**Conceptualization:** Kerui Pan, Xueping Li, Shuoming Wu.

**Data curation:** Kerui Pan, Sisi Wang, Xueping Li, Shuoming Wu.

**Formal analysis:** Kerui Pan, Sisi Wang, Xueping Li, Shuoming Wu.

**Funding acquisition:** Sisi Wang, Xueping Li, Shuoming Wu.

**Investigation:** Kerui Pan, Sisi Wang, Xueping Li, Shuoming Wu.

**Resources:** Kerui Pan, Sisi Wang, Xueping Li.

**Software:** Kerui Pan, Xueping Li, Shuoming Wu.

**Supervision:** Kerui Pan, Xueping Li, Shuoming Wu.

**Validation:** Kerui Pan, Sisi Wang, Shuoming Wu.

**Visualization:** Kerui Pan, Sisi Wang, Xueping Li, Shuoming Wu.

**Writing – original draft:** Kerui Pan, Sisi Wang.

**Writing – review & editing:** Kerui Pan, Sisi Wang, Xueping Li, Shuoming Wu.

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
