## [Decision Letter · Decision Letter 0]

14 May 2024

PONE-D-24-11718Efficacy and safety of ultrasound-guided radiofrequency ablation combined with transhepatic artery embolization chemotherapy for hepatocellular carcinoma: a meta-analysisPLOS ONE

Dear Dr. Wu,

Thank you for submitting your manuscript to PLOS ONE. After careful consideration, we feel that it has merit but does not fully meet PLOS ONE’s publication criteria as it currently stands. Therefore, we invite you to submit a revised version of the manuscript that addresses the points raised during the review process.

We look forward to receiving your revised manuscript.

Kind regards,

Ashraf Elbahrawy

Academic Editor

PLOS ONE

2. We note that your Data Availability Statement is currently as follows: [The authors confirm that the data supporting the findings of this study are available within the article and its supplementary materials.]

3. PLOS requires an ORCID iD for the corresponding author in Editorial Manager on papers submitted after December 6th, 2016. Please ensure that you have an ORCID iD and that it is validated in Editorial Manager. To do this, go to ‘Update my Information’ (in the upper left-hand corner of the main menu), and click on the Fetch/Validate link next to the ORCID field. This will take you to the ORCID site and allow you to create a new iD or authenticate a pre-existing iD in Editorial Manager. Please see the following video for instructions on linking an ORCID iD to your Editorial Manager account: https://www.youtube.com/watch?v=_xcclfuvtxQ.

6. We are unable to open your Supporting Information file [Supplementary Material .docx]. Please kindly revise as necessary and re-upload.

Reviewers' comments:

Reviewer's Responses to Questions

**Comments to the Author**

1. Is the manuscript technically sound, and do the data support the conclusions?

Reviewer #1: Yes

Reviewer #2: Yes

2. Has the statistical analysis been performed appropriately and rigorously? 

Reviewer #1: Yes

Reviewer #2: Yes

3. Have the authors made all data underlying the findings in their manuscript fully available?

Reviewer #1: No

Reviewer #2: Yes

4. Is the manuscript presented in an intelligible fashion and written in standard English?

Reviewer #1: Yes

Reviewer #2: No

5. Review Comments to the Author

Reviewer #1: Dear authors, thanks for this great, interresting and important work.

- The methodology is good , but I think, it may need more details. Like:

1. How combined TACE and RFA were done (which one was the first modality and what was the time interval between them). And what is the type of chemotherapy in each TACE study.

2. Which staging system you used for HCC (BCLC or WHO) and the comparison of staging between each groups (combined TACE and RFA group vs TACE only group).

3. You didn't mention anything about previous treated patients (were they excluded?)

- I think, it would be good if you made a hint about cost benefit ratio (if possible).

At the end, it is very benificial work and i am looking forward seeing your reply.

Reviewer #2: The language in submitted article has many errors and repeated sentences carrying the same meaning. Also there are many typographical l errors that should be corrected at revision.

I can not understand the dead line ( a search deadline of 14 March 1, 2024) please clarify

Definition of cirrhosis is not correct (cirrhosis (chronic liver damage caused by chronic liver damage caused by fibrosis) it should include regenerating nodules to differentiate cirrhosis from fibrosis.

Subtitles should be clear and obvious in different size

6. PLOS authors have the option to publish the peer review history of their article (what does this mean?). If published, this will include your full peer review and any attached files.

Reviewer #1: **Yes: **Elfayoumie M

Reviewer #2: **Yes: **Ali Madian

---

## [Author Response · Author response to Decision Letter 0]

15 May 2024

Dear editor and dear reviewers

Re Manuscript ID (PONE-D-24-11718) entitled " Efficacy and safety of ultrasound-guided radiofrequency ablation combined with transhepatic artery embolization chemotherapy for hepatocellular carcinoma: a meta-analysis". Thank you for your letter and for the reviewer’s comments. Those comments are all valuable and very helpful for revising and improving our paper, as well as the important guiding significant to our research. We have studies comments carefully and have made correction which we hope meet with approval. Revised portion are marked in red in the paper. The main correction in the paper and responds to the reviewer’s are as flowing:

Reviewer #1: Dear authors, thanks for this great, interresting and important work.

- The methodology is good , but I think, it may need more details. Like:

1. How combined TACE and RFA were done (which one was the first modality and what was the time interval between them). And what is the type of chemotherapy in each TACE study.

Responds: Many thanks to the reviewers for their valuable comments, through our careful reading of the articles, we found that most of the included articles performed TACE followed by radiofrequency ablation, but the intervals between the two interventions were not the same in each article, and we added the method of radiofrequency ablation used in each article in Table 1 Literature Characterization Table.

2. Which staging system you used for HCC (BCLC or WHO) and the comparison of staging between each groups (combined TACE and RFA group vs TACE only group).

Responds: Many thanks to the reviewers for their valuable comments, most of the studies we included fall under WHO staging. And some of the studies were not grouped according to interventions for staging, but more directly gave me the staging for all the included population.

3. You didn't mention anything about previous treated patients (were they excluded?)

- I think, it would be good if you made a hint about cost benefit ratio (if possible).

Responds: Many thanks to the reviewers for their questions, which will be a great enhancement to our article, and indeed people who have previously received other treatments will be excluded by us, which we have added to the exclusion criteria.

At the end, it is very benificial work and i am looking forward seeing your reply.

Responds: We strongly agree with the reviewers' comments, but the included articles provided less information for us to cost-effect analyze, but we will dedicate a future study to this cost-effect analysis.

Reviewer #2: The language in submitted article has many errors and repeated sentences carrying the same meaning. Also there are many typographical l errors that should be corrected at revision.

I can not understand the dead line ( a search deadline of 14 March 1, 2024) please clarify

Responds: We are very sorry that we didn't explain it clearly, it means that our retrieval date is March 14, 2024.

Definition of cirrhosis is not correct (cirrhosis (chronic liver damage caused by chronic liver damage caused by fibrosis) it should include regenerating nodules to differentiate cirrhosis from fibrosis.

Responds: Many thanks to the reviewers for pointing out the problems, which we have revised.

Subtitles should be clear and obvious in different size

Responds: Many thanks to the reviewers for pointing out the problems, which we have revised.

---

## [Decision Letter · Decision Letter 1]

10 Jun 2024

Efficacy and safety of ultrasound-guided radiofrequency ablation combined with transhepatic artery embolization chemotherapy for hepatocellular carcinoma: a meta-analysis

PONE-D-24-11718R1

Dear Dr. Wu,

We’re pleased to inform you that your manuscript has been judged scientifically suitable for publication and will be formally accepted for publication once it meets all outstanding technical requirements.

Kind regards,

Ashraf Elbahrawy

Academic Editor

PLOS ONE

Additional Editor Comments (optional):

Reviewers' comments:

Reviewer's Responses to Questions

**Comments to the Author**

1. If the authors have adequately addressed your comments raised in a previous round of review and you feel that this manuscript is now acceptable for publication, you may indicate that here to bypass the “Comments to the Author” section, enter your conflict of interest statement in the “Confidential to Editor” section, and submit your "Accept" recommendation.

Reviewer #1: All comments have been addressed

Reviewer #2: All comments have been addressed

2. Is the manuscript technically sound, and do the data support the conclusions?

Reviewer #1: Yes

Reviewer #2: Yes

3. Has the statistical analysis been performed appropriately and rigorously? 

Reviewer #1: Yes

Reviewer #2: Yes

4. Have the authors made all data underlying the findings in their manuscript fully available?

Reviewer #1: Yes

Reviewer #2: Yes

5. Is the manuscript presented in an intelligible fashion and written in standard English?

Reviewer #1: Yes

Reviewer #2: Yes

6. Review Comments to the Author

Reviewer #1: (No Response)

Reviewer #2: The authors have adequately addressed previous comments. The manuscript presented in an intelligible fashion and written in standard English

7. PLOS authors have the option to publish the peer review history of their article (what does this mean?). If published, this will include your full peer review and any attached files.

Reviewer #1: **Yes: **M. S. El-Fayoumie

Reviewer #2: **Yes: **Ali Madian

---

## [Editor Report · Acceptance letter]

30 Jul 2024

PONE-D-24-11718R1 

PLOS ONE

Dear Dr. Wu, 

I'm pleased to inform you that your manuscript has been deemed suitable for publication in PLOS ONE. Congratulations! Your manuscript is now being handed over to our production team.

Kind regards, 

on behalf of

Prof. Ashraf Elbahrawy 

Academic Editor

PLOS ONE